# Task-Robust Pre-Training for Worst-Case Downstream Adaptation

**Jianghui Wang**[*] **Yang Chen**[*], **Xingyu Xie, Cong Fang**[†] **Zhouchen Lin**[†]

School of Intelligence Science and Technology, Peking University

jianghuiwang.ai@gmail.com, {yangchen, xyxie, fangcong, zlin}@pku.edu.cn

## Abstract

Pre-training has achieved remarkable success when transferred to downstream tasks. In machine learning, we care about not only the good performance of a model but also its behavior under reasonable shifts of condition. The same philosophy holds when pre-training a foundation model. However, the foundation model may not uniformly behave well for a series of related downstream tasks. This happens, for example, when conducting mask recovery regression where the recovery ability or the training instances diverge like pattern features are extracted dominantly on pre-training, but semantic features are also required on a downstream task. This paper considers pre-training a model that guarantees a uniformly good performance over the downstream tasks. We call this goal as *downstream-task robustness*. Our method first separates the upstream task into several representative ones and applies a simple minimax loss for pre-training. We then design an efficient algorithm to solve the minimax loss and prove its convergence in the convex setting. In the experiments, we show both on large-scale natural language processing and computer vision datasets our method increases the metrics on worse-case downstream tasks. Additionally, some theoretical explanations for why our loss is beneficial are provided. Specifically, we show fewer samples are inherently required for the most challenging downstream task in some cases.

## 1  Introduction

The rapid development of machine learning is promoting a shift in the learning paradigm, where one first trains a very large model, often called a foundation model, with massive data, and then adapts it to desired tasks using much less data. The hope is to obtain a model that serves as an infrastructure and is transferable to a wide range of tasks. Pre-training plays the role of an engine to acquire the foundation model. Typical pre-training methods are to minimize the average expected risks of the upstream tasks. Such pre-trained models can achieve good performance for a lot of downstream tasks but may fail in some hard cases. For example, a vision pre-trained model for animal and plant recognition may work well for typical characteristics species but fail when identifying mimicry animals and plants; a common green mantis can be correctly recognized as a mantis, while an orchid mantis might be falsely classified as an orchid.

In machine learning, one cares about not only the good performance of a model but also its behavior under reasonable shifts of conditions. The same philosophy holds for pre-training a foundation model. To guarantee a uniformly good performance over a series of tasks, one has to consider the robustness of pre-training. We call this *downstream-task robustness*. The aim is to develop a pre-training method to train the foundation model that admits a good adaptation performance over a series of downstream

---

[*]Equal contribution.
[†]Corresponding author.

37th Conference on Neural Information Processing Systems (NeurIPS 2023).

tasks. It is crucial to achieving the downstream-task robustness for pre-training: (i) safety is critical for some applications, such as deep learning systems in medicine and finance; (ii) our goal of the foundation model requires reliably good performance on all downstream tasks.

In recent times, popular large-scale models like ChatGPT [46] have also faced safety issues, sparking discussions among various stakeholders [8]. The concerns largely stem from the potential misuse or unintended behavior of the model in real-world applications. Some argue that the vast and diverse knowledge base of these models, coupled with their ability to generate human-like text, could be leveraged for malicious purposes. Others worry about the potential of the model to generate inappropriate or harmful content [47].

Addressing these safety concerns is crucial, particularly in the context of pre-training foundation models that are intended for downstream tasks. An initial way to mitigate these safety issues is to consider downstream-task robustness. By focusing on downstream-task robustness, we expect that the models are resilient to perturbations in the input data, thereby reducing their susceptibility to adversarial attacks or misuse. This approach can help in maintaining consistent performance across a range of tasks and so enhance the safety and reliability of the model.

In recent years, the concept of Distributionally Robust Optimization (DRO) [6, 44, 36] has attracted wide attention among theorists and practitioners. Most DRO frameworks [55, 23, 22] consider training a parameterized model that minimizes the worst-case expectation loss over the data from a family of probability distributions. Downstream-task robustness can be considered a generalization of DRO. The destination of downstream-task robustness is to guarantee good worst-case performance for a series of downstream adaptations.

This paper proposes a pre-training method as a starting point for downstream-task robustness. To take a step forward, our method considers learning several upstream tasks. The choice of how to design the upstream tasks allows us to incorporate prior knowledge of the domain and problems. For example, in language models, we can design upstream tasks by masking different types of words; how we generate such upstream tasks by grouping samples reflects our prior knowledge of the natural language. Then instead of minimizing the average expected risk of the upstream tasks, we minimize the worst-case expected risk. We also introduce a simple but practical algorithm called softmax weighted gradient descent to pre-train the model. We prove the algorithm's convergence in the ideal convex setting and show its effectiveness in our empirical study.

We consider the application of the framework in two experiments — Part-of-Speech masked language models in section 4.1 and multi-modal masked image models in section 4.2. We first pre-train a foundation model with the proposed task-robust pre-training method on multiple upstream tasks generated by different masks and adapt the foundation model for downstream tasks. Compared with the average expected risk minimization, our method achieves better worst-case performance and comparable average performance on all downstream tasks.

We also explain why our framework can benefit downstream-task robustness. Specifically, by simplifying the model-task relationship, we show fewer samples are needed for the hardest downstream task. The key intuition is that proper worse-case training for upstream tasks leads to an initiation close to the solution for the worst-case downstream tasks, thus reducing the downstream burden.

The contributions of our study can be primarily encapsulated within three key areas: (i) The introduction of the concept of task-robust pretraining, a novel theoretical framework that holds significant potential for future research. (ii) The provision of a simple yet efficacious method for task-robust pretraining, accompanied by a comprehensive exposition of its theoretical feasibility. (iii) A series of empirical validations across multiple domains, substantiating the effectiveness of our methodology.

## 2   Setup and Methodology

Consider a traditional machine learning task where the data $z \in \mathbb{Z}$ follows an underlying distribution $P$. Given a model, a parameter space $\Theta \subset \mathbb{R}^d$, a loss function $\ell : \Theta \times \mathbb{Z} \mapsto \mathbb{R}_+$, the goal is to find the optimal parameter $\theta^*$ for the model such that $\theta^* = \arg\min_{\theta \in \Theta} \mathbb{E}_{z \sim P} [\ell(\theta, z)]$. The classic empirical risk minimization (ERM) tackles the problem by first collecting i.i.d. training data $\{z_i\}_{i=1}^N$

from $P$ and then finding a parameter $\hat{\theta}_{\text{ERM}}$ via minimizing the empirical risk:

$$\hat{\theta}_{\text{ERM}} := \arg\min_{\theta \in \Theta} \frac{1}{n} \sum_{i=1}^{N} \ell(\theta, z_i). \tag{1}$$

In statistical learning theory, it is well-known that under mild conditions (such as the VC dimension of the model is bounded above), $\hat{\theta}_{\text{ERM}}$ is a good approximation of $\theta^*$ in the sense that with high probability at least $1 - \delta$ ( $0 < \delta \ll 1$),

$$\mathbb{E}_{z \sim P}\left[\ell\left(\hat{\theta}_{\text{ERM}}, z\right)\right] - \min_{\theta \in \Theta} \mathbb{E}_{z \sim P}[\ell(\theta, z)] \leq \epsilon, \tag{2}$$

when the number of training samples $N$ is sufficiently large. We simply denote the training data requirement by $N(\epsilon, \delta)$.

In machine learning, a foundation model is trained and then adapted for each downstream task. The adaptation process involves initializing a downstream model with the pre-trained foundation model's parameters and then training on the downstream task. Denote the parameter of the foundation model by $\theta_{\text{foundation}}$. Let $\Lambda$ be the downstream task space. The goal is to find an initial parameter $\theta_{\text{foundation}}$ that enables fast adaptations for each downstream task $\lambda \in \Lambda$. Different downstream tasks are characterized by different loss functions with a shared data distribution [3]. We use the foundation model's parameters $\theta_{\text{foundation}}$ as initialization to find an approximately optimal parameter $\hat{\theta}_{\lambda,\text{ERM}}$ by ERM. We study the sample complexity required to guarantee a good approximate solution with high probability while considering the effect of initialization. For the task $\lambda$ and the model initialized by $\theta_{\text{foundation}}$, denote the number of samples required to find an $\epsilon$-approximately optimal parameter by ERM with probability at least $1 - \delta$ as $N_\lambda(\theta_{\text{foundation}}, \epsilon, \delta)$. The aim is to find the optimal initialization that minimizes the worst-case sample complexity required to find an approximately optimal parameter for all tasks, i.e.,

$$\theta_{\text{foundation}}^* := \arg\min_{\theta_{\text{foundation}} \in \Theta} \max_{\lambda \in \Lambda} N_\lambda(\theta_{\text{foundation}}, \epsilon, \delta). \tag{3}$$

Directly training for the optimal worst-case initialization is generally infeasible or computationally expensive. Pre-training provides a feasible alternative $\theta_{\text{pre-train}}^*$ for $\theta_{\text{foundation}}^*$ by training for available surrogate upstream tasks. When the upstream tasks are related to the downstream tasks, the pre-trained parameter can lead to lower initial expected risks and accelerate downstream training. For example, if we pre-train a model on generated upstream tasks of reconstructing images corrupted by different masks, the pre-trained model can learn some prior knowledge for general vision tasks; with the pre-trained parameter as the initialization, we can accelerate the training process of downstream vision tasks such as image classification and object detection [13]. Consider there are $T$ representative upstream tasks. Denote the loss function of the task $t$ as $\ell_t$. A typical choice of the pre-trained parameter is the minimizer of the average expected risk over the $T$ upstream tasks [43, 18, 29], i.e.,

$$\theta_{\text{average}}^* := \arg\min_{\theta \in \Theta} \frac{1}{T} \sum_{t=1}^{T} \mathbb{E}_{z \sim P}[\ell_t(\theta, z)]. \tag{4}$$

However, minimizing the average expected risk over upstream tasks may neglect extreme cases and lead to limited benefit for some downstream tasks.

To alleviate the aforementioned issue of the average expected risk minimization, we propose to use the minimizer of the worst-case expected risks over the upstream tasks, i.e.,

$$\theta_{\text{max}}^* := \arg\min_{\theta \in \Theta} \max_{t \in [T]} \mathbb{E}_{z \sim P}[\ell_t(\theta, z)], \tag{5}$$

as the initial parameter, where $[m]$ denotes the set $\{1, \ldots, m\}$. We show that $\theta_{\text{max}}^*$ is a better choice than $\theta_{\text{average}}^*$ in terms of downstream-task robustness.

---

[3]Data distributions vary for different tasks can be modelled by incorporating weighting functions from the data distributions to the loss functions. Consider a task where the data distribution is $P_t$ and $\frac{dP_t}{dP}(z) > 0$ for all $z \in \text{supp}(P_t)$, the expected loss is $\mathbb{E}_{z \sim P_t}[\ell(\theta, z)]$. The expected loss can still be rewritten as $\mathbb{E}_{z \sim P}[\ell_t(\theta, z)]$, where $\ell_t(\theta, z) = \frac{dP_t}{dP}(z)\ell(\theta, z)$.

# 3 Algorithm

Recall that our minimax pre-training method is to minimize the worst-case expected risks over the upstream tasks, i.e.,

$$\min_{\theta \in \Theta} \max_{t \in [T]} \mathbb{E}_{z \sim P} \left[ \ell_t \left( \theta, z \right) \right] \tag{6}$$

There is extensive literature on minimax optimization. The minimax optimization algorithms can be generally classified into two types: (i) minimization for the maximum function $\max_{t \in [T]} \mathbb{E}_{z \sim P} \left[ \ell_t \left( \theta, z \right) \right]$ [12, 4, 60, 28], and (ii) direct minimax optimization for the objective [35, 59, 45, 37, 38]. We introduce a new simple optimization algorithm called softmax weighted gradient descent (Algorithm 1) that we find is very practical to pre-train the model. The algorithm can be roughly seen as the first type. It is an adaptation of the classic subgradient descent for minimizing the maximum function to enable its practical use in pre-training. Concretely, in one update, we take a descent step at the current point $\theta$ along the direction of the gradient weighted by softmax-type weights, i.e., $\sum_{t=1}^{T} w_{\alpha, t} \left( \theta \right) \nabla_\theta \mathbb{E}_{z \sim P} \left[ \ell_t \left( \theta, z \right) \right]$, where

$$w_{\alpha, t} \left( \theta \right) := \frac{\exp \left( \alpha \mathbb{E}_{z \sim P} \left[ \ell_t \left( \theta, z \right) \right] \right)}{\sum_{t'=1}^{T} \exp \left( \alpha \mathbb{E}_{z \sim P} \left[ \ell_{t'} \left( \theta, z \right) \right] \right)}, \tag{7}$$

and $\alpha > 0$ is a hyperparameter. (In practice, we use estimations for the expected risks and the gradients on minibatch samples.)

The motivation behind softmax weighted gradient descent is to use the softmax weighted gradient to approximate the subgradient in the classic subgradient descent for the minimax optimization of (6). As the softmax weighted gradient descent algorithm optimizes for the minimax loss directly, it can achieve better worst-case loss than other pre-training methods. One advantage of the softmax approximation is that it avoids the non-differentiability caused by the maximum operator via the softmax approximation, making the algorithm easily implementable for pre-training applications. Also, as the softmax weighted gradient descent includes only single weighted gradient step in each update, it has computational efficiency comparable to gradient descent in deep learning. In contrast, standard minimax algorithms often cost several times gradient oracles in single step. Moreover, our algorithm can be directly combined with commonly-used optimization tricks in deep learning, such as momentum and adaptive learning rates. In our experiments, we observe that the algorithm with a simple implementation achieves better worst-case errors in various real-world tasks than a number of benchmark pre-training algorithms.

---

**Algorithm 1** Softmax Weighted Gradient Descent

---

**Input:** Step sizes $\{\eta_k\}_{k=1}^{K-1}$, softmax hyperparameters $\{\alpha_k\}_{k=0}^{K-1}$ and an initial parameter $\theta_0$;
**for** $k = 1, \ldots, K - 1$ **do**
    Compute the softmax weights $\{w_{\alpha_k, t} \left( \theta_{k-1} \right)\}_{t=1}^{T}$ as in (7);
    Update the parameter as $\theta_k \leftarrow \theta_{k-1} - \eta_k \sum_{t=1}^{T} w_{\alpha_k, t} \left( \theta_{k-1} \right) \nabla_\theta \mathbb{E}_{z \sim P} \left[ \ell_t \left( \theta_{k-1}, z \right) \right]$;
**end for**

---

For completeness, we also provide some convergence analysis for our proposed algorithm. We consider a relatively basic setting where for all $t \in [T]$, the loss function $\ell(\cdot, z)$ is convex and $L'$-Lipschitz continuous for any fixed $z \in \mathbb{Z}$ Intuitively, when the hyperparameter $\alpha_k$ is sufficiently large, the function $\sum_{t=1}^{T} w_{\alpha_k, t} \left( \theta_k \right) \mathbb{E}_{z \sim P} \left[ \ell_t \left( \theta, z \right) \right]$ is a good differentiable approximation for the objective $\mathbb{E}_{z \sim P} \left[ \ell_t \left( \theta, z \right) \right]$ Softmax weighted gradient descent can be roughly seen as a remedy for non-differentiability in subgradient descent, at the expense of controllable approximation errors. We show in Theorem 3.1 that Algorithm 1 can achieve a convergence rate $O \left( \frac{1}{\sqrt{K}} \right)$ if the hyperparameter $\alpha_k$ is as large as $\tilde{O}(\sqrt{k})$. This result is comparable to the standard convergence rate $O \left( \frac{1}{\sqrt{K}} \right)$ of subgradient descent [10, Chapter 3].

**Theorem 3.1.** *Suppose that for all $t \in [T]$ the loss function $\ell_t(\cdot, z)$ is convex, $L'$-Lipschitz continuous and bounded by $B$ for all $\theta \in \Theta$ and any fixed $z \in \mathbb{Z}$. Denote the optimal solution of (6) as $\theta^*$ and the distance $\|\theta_0 - \theta^*\|$ as $R_0$. If the step size $\eta_k = \eta = \frac{R_0}{L' \sqrt{K}}$ and the softmax hyperparameter $\alpha_k \geq \frac{4\sqrt{k+1}}{R_0 L'} \log \frac{4TB\sqrt{k+1}}{R_0 L'}$ for all $k = 0, \ldots, K - 1$, the average $\bar{\theta}_K$ of the iteration points in*

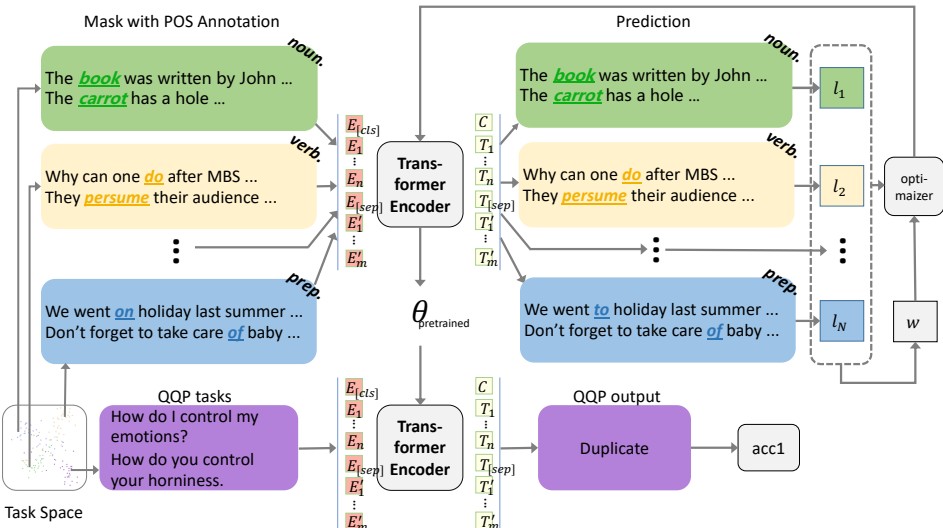

Figure 1: **Part-of-Speech Mask BERT.** We first sample the datasets from the task space and group them according to different Part-of-Speech types, and then recover the predicted sentence by a BERT encoder. The optimizer selects the most challenging task and then updates the model's weight through a minimax layer.

*Algorithm 1, i.e., $\bar{\theta}_K = \frac{1}{K}\sum_{k=0}^{K-1} \theta_k$, satisfies*

$$\max_{t\in[T]} \mathbb{E}_{z\sim P}\left[\ell_1\right] - \min_{\theta\in\Theta}\max_{t\in[T]} \mathbb{E}_{z\sim P}\left[\ell_2\right] \leq \frac{2R_0 L'}{\sqrt{K}}, \tag{8}$$

*where $\ell_1 = \ell_t\left(\bar{\theta}_K, z\right)$ and $\ell_2 = \ell_t\left(\theta, z\right)$.*

*Remark* 3.2. The convexity assumption on the loss functions is an oversimplification in deep learning. However, for some cases such as the neural tangent kernel [32], the deep neural networks exhibit properties similar to convexity. In our experiments with non-convex models, we also observe that the algorithm behaves well.

*Remark* 3.3. The above analysis requires increasing softmax hyperparameters $\{\alpha_k\}_{k=0}^{K-1}$, i.e., $\alpha_k = \tilde{O}\left(\sqrt{k}\right)$ to guarantee the convergence rate. In our experiments, however, we find that constant softmax hyperparameters, or more concretely $\alpha_k = 1$ for all $k = 0, \ldots, K-1$, work well for most problems. We attribute these phenomena to some properties of deep neural networks, which are left for future exploration.

# 4 Experiments

In this section, we subject our methods to rigorous testing through two experiments, each encompassing tasks germane to the fields of Natural Language Processing (NLP) and Computer Vision (CV). A minimalist design approach was adopted for both the models and the tasks to demonstrate the universality of our design across a broad spectrum of model tasks. We extended the functionalities of BERT and MAE, thereby constructing Part-of-Speech Mask BERT (PoS-BERT) and Multi-Modal Mask MAE (MM-MAE), respectively.

## 4.1 NLP Scenario: Part-of-Speech Mask BERT

### 4.1.1 Model and Settings

**Architectures** An overview of the Part-of-Speech Mask BERT model is shown in Figure 1. PoS-BERT model first samples the datasets from the task space and groups them according to different Part-of-Speech types. The loss function term is calculated separately for each data group entering the BERT encoder. The loss term with the highest weight is selected to enter the optimizer through a minimax layer. Finally, we run experiments on downstream tasks to compare our minimax task

Table 1: Results on GLUE. The "Averages" are obtained from GLUE leaderboard. F1 scores are reported for QQP and MRPC. , spearman correlations are reported for STS-B, Matthews correlations are reported for CoLA, and accuracy scores are reported for the other tasks.

| Model | Task-Balancing | MNLI | QQP | QNLI | SST-2 | CoLA | STS-B | MRPC | RTE | Avg. |
|---|---|---|---|---|---|---|---|---|---|---|
| BERT | - | 84.6 | 71.2 | **90.5** | **93.5** | 52.2 | **85.8** | **88.9** | 66.4 | 79.6 |
| PoS-BERT | None | 84.9 | 72.6 | 89.1 | 90.8 | 54.4 | 83.6 | 88.1 | 68.2 | 79.3 |
| | Minimax (Ours) | **85.6** | **76.9** | 88.6 | 91.3 | **61.4** | 84.2 | 88.2 | **70.7** | **81.4 (+1.8)** |

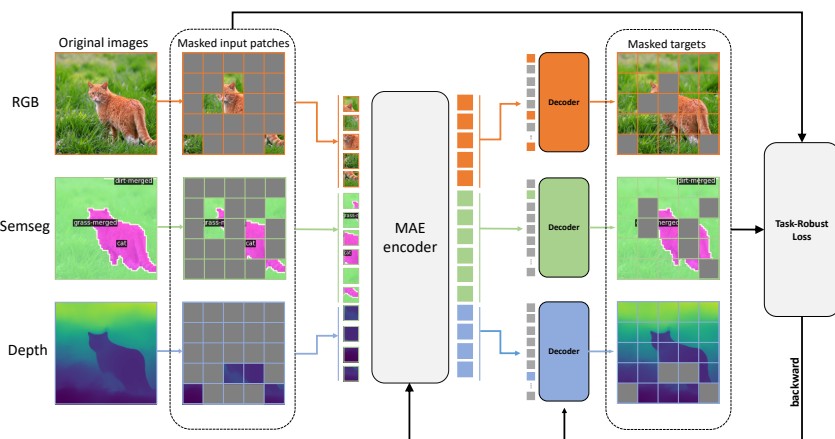

Figure 2: **Multi-Modal Mask MAE:** Randomly sampled patches from multiple modalities are projected to tokens. Task-specific decoders reconstruct the masked-out patches by first performing a cross-attention step from queries to the encoded tokens.

balancing learning algorithm with other methods to verify our theory. We follow the common practice to design the feature representations for masked language modeling and next-sentence prediction.

**Tasks and Datasets** During pre-training, Part-of-Speech Mask BERT has two objectives: masked language modeling and next-sentence prediction. We define 9 task categories that recover different parts of speech-type words on masked language modeling: (1)verb, (2) noun, (3) adjective, (4) determiner, (5) adverb, (6) pronoun, (7) preposition, (8) conjunction, (9) interjection. Following previous work [18, 48, 40] , we evaluate our pre-trained models on downstream tasks using the GLUE [64] benchmarks. Downstream tasks we fine-tuned include MNLI, QQP, QNLI, SST-2, CoLA, STS-B, MRPC, and RTE. By swapping out the appropriate inputs and outputs, Part-of-Speech Mask BERT can model many downstream tasks and has a unified way to handle the tasks that involve single text and text pairs. After setting the masks, we utilize Natural Language Toolkitannotate (NLTK) [7] to pseudo-label the words with POS annotations.

### 4.1.2 Quantitative Result

Quantitative results are presented in Table 1. Our model obtains comparable results on GLUE tasks. PoS-BERT with minimax task-balancing outperforms on half tasks by a substantial margin and obtains a 1.8% average score improvement over BERT. As for the most challenging training task, CoLA, which has the lowest accuracy on BERT, our method gets a 9.2% improvement which is a significant boost among downstream tasks. Benefiting from task-robust grouping, on QQP and RTE tasks, our method outperforms the original BERT by 5.7 F1-score and 4.3% accuracy, respectively. Our method also shows superiority on the challenging downstream task, MNLI, by achieving a 1.1% higher matched accuracy. As compensation for working better on the more challenging tasks, our method loses little correctness on some downstream tasks that already transfer well. On QNLI, SST-2, STS-B, and MRPC, our results are lower than that of the original BERT model by a margin of 1% accuracy, 1.9% accuracy, 1.6 spearson correlation, and 0.7 F1-score. The empirical result shows that, with our task-robust pre-training strategy, the downstream tasks perform on the whole, especially those tricky tasks. We expect future work to further improve these results by incorporating more sophisticated multi-task and grouping procedures.

Table 2: Comparison between task-robust method and other task-balancing methods on ImageNet1K and ImageNetS50 pre-training. Our methodology demonstrates superior performance across the majority of downstream tasks, particularly excelling in the most challenging among them.

| Pre-training | | | Downstream | | |
|---|---|---|---|---|---|
| Data | Epoch | Task-Balancing | Cls. (Top-1 Acc. %) | Semseg. (mIoU) | Depth. ($\delta_1$ Acc. %) |
| ImageNetS50 | 800 | None | 92.2 | 51.9 | 52.1 |
| | | Uncertainty [34] | 92.6 | 54.5 | 70.2 |
| | | GradNorm [16] | 93.0 | 56.5 | 65.8 |
| | | DWA [39] | **93.4** | 52.7 | 65.7 |
| | | Minimax(Ours) | 91.8 | **61.5** | **74.1** |
| ImageNet1K | 400 | Uncertainty | **82.6** | 48.9 | 85.2 |
| | | Minimax(Ours) | 82.3 | **50.1** | **85.3** |
| | 1600 | Uncertainty | **83.3** | 52.0 | 86.4 |
| | | Minimax(Ours) | 83.0 | **53.2** | **86.8** |

Table 3: A qualitative comparison between task balancing techniques. $T$ representing the computation cost when no additional task-balancing techniques are employed.

| Method | Balance Magnitude | Balance Learning | Grads Required | No Extra Tuning | FLOPs | Motivation |
|---|---|---|---|---|---|---|
| None | ✓ | | | ✓ | $T$ | / |
| Uncertainty | ✓ | | | ✓ | $2T$ | Homoscedastic uncertainty |
| Gradnorm | ✓ | ✓ | ✓ | ✓ | $4T$ | Balance learning and magnitudes |
| DWA | | ✓ | | | $3T$ | Balance learning |
| Minimax (Ours) | ✓ | | | ✓ | $2T$ | Task robust |

## 4.2 CV Scenario: Multi-Modal Mask MAE

### 4.2.1 Model and Settings

**Architectures** An overview of MM-MAE is shown in Figure 4.1.2. Multi-Modal Mask MAE contains three encoders, each of which processes different modalities of one image. During pre-training, we try to recover each modality from its masked tokens. Each modality is divided into $16 \times 16$ patches and then tokenize the patches with modality-dependent linear projections. Projected patches are concatenated into a sequence of tokens and given as input to the same transformer encoder. We also add a global token with 2D sine-cosine positional embeddings. Each task owns a specialized decoder, and the computational cost of decoders scales linearly with the number of tasks.

**Tasks and Datasets** We select two datasets with different scales, ImageNet1K [53] and ImageNetS50 [25], to conduct unsupervised training upstream to see whether the minimax pre-training method can help the downstream tasks with poor performance. The classification task is evaluated on the validation part of the original dataset, while the semantic segmentation and depth estimation tasks are validated on the NYUv2 dataset [56] by fine-tuning. Due to the absence of a sizeable multi-task dataset with aligned task images [19, 2] we generate pseudo-labels on ImageNet and ImageNetS50 with GPT-3 and Mask2Former.

### 4.2.2 Quantitative Result

**Classification tasks** The quantitative results are presented in Table 2. We evaluate our models and baseline by fine-tuning them on the supervised ImageNetS50 and ImageNet1K. We fine-tune our models for 100 epochs and report the top-1 validation accuracy. The result shows a tiny gap between our method and the average method in the classification task. Classification tasks are regarded as the least challenging task category of the three. Cause different downstream tasks have different optimal parameter requirements, this gap is unavoidable. After the pre-training of the model reaches a specific step, the training weight of the classification tasks will continue to decrease.

**Semantic segmentation tasks** We further evaluate our models on semantic segmentation tasks on the NYUv2 dataset. We report the mean intersection over the union (mIoU) metric. Notice that semantic segmentation is the most challenging task of these downstream transfers. Our method

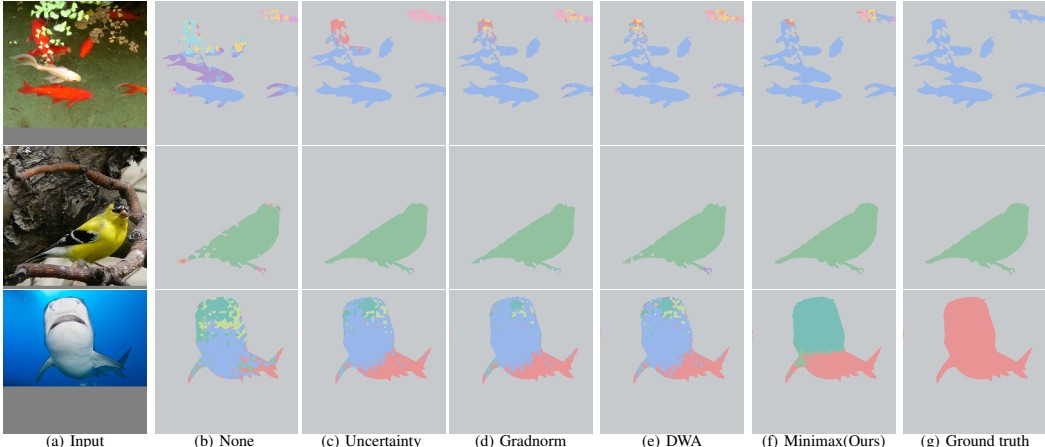

| (a) Input | (b) None | (c) Uncertainty | (d) Gradnorm | (e) DWA | (f) Minimax(Ours) | (g) Ground truth |

Figure 3: Comparative intermediate results. The first and final columns represent the image input and ground truth, respectively, while the intermediary images depict the intermediate results yielded by various task-balancing methodologies. Distinct colors correspond to the prediction of different objects. Our approach ensures robustness in downstream tasks.

benefits more than the average loss model from pseudo-labeled modalities as input. In particular, the correctness is improved by 9.6% on ImageNetS50 pre-training. With the progress of model training, our task-robust loss forces the model to improve poorly trained semantic segmentation tasks by increasing the training weight. The following section 5 will explain why a simple strategy can significantly help worst-case downstream tasks.

**Depth evaluate tasks** For depth estimation, we use NYUv2 . We report $\delta_1$ on the NUYv2 test set, showing the percentage of pixels $p$ with error max $\{\frac{\hat{y_p}}{y_p}, \frac{y_p}{\hat{y_p}}\}$ less than 1.25 [20]. According to Table 2, the accuracy is improved by 3.9% on ImageNetS50 pre-training with the help of the downstream-task robustness loss function. The depth estimation task in the same data volume has a higher tolerance for prediction errors per pixel than semantic segmentation. However, it is still more complicated than the classification task, which only predicts the image once. After the classification task is well-trained, the depth estimation task will benefit from our strategy in the subsequent training.

### 4.3 Qualitative Comparison

Table 2 delineates several strategies designed to equilibrate the contribution of each task during the training of a multi-task network. For a qualitative comparison of these methods, refer to Table 3. We appraise these strategies based on several criteria [61]. An overview of our examination suggests that our approach achieves a synergistic blend of simplicity, efficiency, and effectiveness.

To facilitate a more intuitive comparison of the differences in results throughout the training process, we undertook downstream tasks in semantic segmentation, comparing the performance of various approaches midway through pre-training (400 epoch, ImageNetS50). As depicted in Figure 3, our methodology exhibits superior performance, even under conditions of insufficient training.

## 5 Explanation

We show why the proposed minimax pre-training method can be more effective than the average expected risk minimization in some cases. We consider a simplification of the model and the task relationship. Such a simplification makes our analysis convenient and intuitive. We assume that for all $t \in [T]$, the function $\ell_t(\cdot, z)$ is $\mu$-strongly-convex, $L$-smooth, and $L'$-Lipschitz continuous for any fixed $z \in \mathbb{Z}$ and the function $\ell_t(\cdot, \cdot) \leq B$ for all $\theta \in \Theta$ and $z \in \mathbb{Z}^4$. Note that pre-training on

---

[4]In fact, it is sufficient to assume that the expectation $\mathbb{E}_z[\ell_t(\cdot, z)]$ is $\mu$-strongly-convex, $L$-smooth, and $L'$-Lipschitz continuous.

irrelevant upstream tasks does little help to the downstream tasks in general. Here, we only discuss the case where the upstream tasks and the downstream tasks are closed related. We ideally assume that the loss functions of the downstream tasks are convex combinations of the loss functions of the upstream tasks, i.e.,

$$\ell_\lambda = \sum_{t=1}^{T} \lambda_t \ell_t, \text{ for all } \lambda \in \Lambda = \Delta_T, \tag{9}$$

where $\Delta_T$ is the $(T-1)$-dimensional probability simplex. We further assume that for each task there exists a parameter such that the expected risk of the task is zero, i.e., $\min_{\theta \in \Theta} \mathbb{E}_{z \sim P} [\ell_t (\theta, z)] = 0$ for all $t \in [T]$.

We first show that in the above setting, the proposed minimax optimization pre-training method can guarantee a better worst-case initial expected risk than the minimization method.

**Proposition 5.1.** *Let $\theta_{max}^*$ and $\theta_{average}^*$ be the pre-trained parameters obtained by minimizing the maximal expected risk and the average expected risk, respectively. Then for the worst-case expected risks of the downstream tasks, we have*

$$\max_{\lambda \in \Lambda} \mathbb{E}_{z \sim P} [\ell_\lambda (\theta_{max}^*, z)] \leq \max_{\lambda \in \Lambda} \mathbb{E}_{z \sim P} [\ell_\lambda (\theta_{average}^*, z)]. \tag{10}$$

*Remark* 5.2. The gap between $\max_{\lambda \in \Lambda} \mathbb{E}_{z \sim P} [\ell_\lambda (\theta_{max}^*, z)]$ and $\max_{\lambda \in \Lambda} \mathbb{E}_{z \sim P} [\ell_\lambda (\theta_{average}^*, z)]$ can be large. We provide an example in the appendix, where the ratio between $\max_{\lambda \in \Lambda} \mathbb{E}_{z \sim P} [\ell_\lambda (\theta_{average}^*, z)]$ and $\max_{\lambda \in \Lambda} \mathbb{E}_{z \sim P} [\ell_\lambda (\theta_{max}^*, z)]$ is as large as $O(T)$.

We then illustrate that a good initialization can serve as an implicit regularization. We suppose that the downstream tasks are trained with gradient descent. (For stochastic gradients with bounded variances, the analysis below also holds for sufficiently small step sizes, within neglectable approximation errors.) For the downstream task $\lambda$, with certain step sizes, the parameters will always be in a subset

$$\Theta_\lambda(\theta_0) = \left\{ \theta \in \Theta \mid \|\theta - \theta_\lambda^*\|^2 \leq \frac{2}{\mu} \mathbb{E}_{z \sim P} [\ell_\lambda (\theta_0, z)] \right\}, \tag{11}$$

where $\theta_0$ is the initial parameter and $\theta_\lambda^* = \arg\min_{\theta \in \Theta} \mathbb{E}_{z \sim P} [\ell_\lambda (\theta, z)]$.

**Proposition 5.3.** *Suppose that a function $f : \mathbb{R}^d \mapsto \mathbb{R}$ is $\mu_f$-strongly-convex and $L_f$-smooth for all $x \in \mathbb{R}^d$ and $x^* \in \arg\min_{x \in \mathbb{R}^d} f(x)$. Let $\{x_k\}_{k=0}^{K-1}$ be the sequence generated by gradient descent with a step size $\eta > 0$, i.e., $x_k = x_{k-1} - \eta \nabla f(x_{k-1})$ for all $k \in [K-1]$. If the step size $\eta \leq \frac{1}{L_f}$, then we have*

$$\|x_k - x^*\|^2 \leq \frac{2}{\mu_f} (f(x_0) - f(x^*)), \tag{12}$$

*for all $k = 0, 1, \ldots, K - 1$.*

By Proposition 5.3, we can deem that the downstream task's parameter space is the subset $\Theta_\lambda(\theta_0)$.

Consider the worst sample complexity to find an $\epsilon$-approximately optimal parameter by ERM within the parameter space $\Theta_\lambda(\theta_0)$ for a downstream task $\lambda \in \Lambda$.

**Theorem 5.4.** *The worst-case sample complexity $\max_{\lambda \in \Lambda} N_\lambda (\theta_0, \epsilon, \delta)$ with initialization $\theta_0$ satisfies*

$$\max_{\lambda \in \Lambda} N_\lambda (\theta_0, \epsilon, \delta) \leq \frac{8dB^2}{\epsilon^2} \log \left( 1 + \frac{16L'}{\epsilon} \sqrt{\frac{2}{\mu} \max_{\lambda \in \Lambda} \mathbb{E}_{z \sim P} [\ell_\lambda (\theta_0, z)]} \right) + \frac{8B^2}{\epsilon^2} \log \frac{2}{\delta}. \tag{13}$$

Theorem 5.4 characterizes the upper bound of the worst-case sample complexity of downstream tasks. If we regard $\epsilon$ and $\delta$ as constants, the worst-case sample complexity with respect to the initialization $\theta_0$ is $O(\log \max_{\lambda \in \Lambda} \mathbb{E}_{z \sim P} [\ell_\lambda (\theta_0, z)])$. Combined with (10), Theorem 5.4 demonstrates that the proposed minimax pre-training procedure implies tighter sample complexity than the average minimization pre-training procedure in the worst case. Though the dependency on the worst-case initial expected risk is logarithmic in the upper bound analysis, we find that the initialization can have an evident effect on the generalization of the downstream tasks in practice. We claim that the upper bound for general cases may not be tight for our deep learning applications. Special structures in applications might lead to tighter bounds for generalization errors, which remains for further study.

# 6  Conclusion

This paper introduces the concept of downstream-task robustness for pre-training, aiming to improve the performance of foundation models across various downstream tasks. As models such as ChatGPT become more prevalent, safety and consistent performance are increasingly important. Our proposed minimax loss for pre-training, validated through extensive experiments, offers a potential solution to enhance the robustness and safety of such models. In the future, we will explore grouping the upstream tasks adaptively. We would say though still in its early stages, the study of downstream-task robustness holds significant promise for the reliable and safe deployment of AI infrastructure.

# 7  Acknowledgments

C. Fang and Z. Lin were supported by National Key R&D Program of China (2022ZD0160301). C. Fang was also supported by the NSF China (No. 62376008) and Wudao Foundation. Z. Lin was also supported by the NSF China (No. 62276004), the major key project of PCL, China (No. PCL2021A12) an Qualcomm.

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

# A Related Work

**Minimax Optimization in Deep Learning.** Minimax optimization has wide application in deep learning, e.g., adversarial robustness [11, 17], distributional robustness [22, 36, 54], adversarial generative models [27], imitation learning [68, 30]. In the field of pre-training, previous work has employed minimax optimization for adversarial robustness [15, 33]. Our work proposes a minimax optimization procedure for pre-training but with a different formulation. While the previous works aims at adversarial robustness, our work focus on the worst-case generalization of downstream tasks.

**Masked language modeling.** Masked language modeling (MLM) was first proposed by [57] as a Cloze task. Adapted as a novel pre-training task, MLM and its autoregressive counterpart ,*e.g.*, BERT [18], GPT [49, 50, 9], and T5 [51], are highly successful methods to deal with NLP problems. MLM first masks out some tokens from the input sentences and then trains the model to retrieve the missing context from the rest of the tokens. These methods have been demonstrated to scale excellently so that various downstream tasks can utilize the pre-trained representations. In particular, BERT is constructed based on the transformer [62] model. After preparing the input samples, an embedding layer and a stack of Transformer layers are followed to conduct the bi-directional semantic modeling. We exploit BERT, the most typical masked language model, as the backbone model to process our ablation experiments.

**Masked image modeling.** Encouraged by transformers, which have gradually become a primary architecture for generic language understanding, ViT [21] later illusion the potential of adopting a pure transformer in image tasks. To generalize better for vision tasks and motivated by the success of BERT [18] in NLP, many recent works propose various masked image prediction methods for pre-training vision models in a self-supervised way. These methods reconstruct the target such as pixels [1, 14, 21, 24, 29, 66], discrete tokens [5, 67], and (deep) features [3, 65]. Notably, the masked autoencoder (MAE) [29] adopts an asymmetric design to allow the large encoder to operate only on unmasked patches and is followed by a lightweight decoder to reconstruct the complete signal from the latent representation along with mask tokens. MultiMAE [2] leverages the efficiency of the MAE approach and extends it to multi-modal and multitask settings. Based on MultiMAE, we apply our approach to increase the transfer capability for downstream tasks.

# B Proofs

## B.1 Convergence Rate of Algorithm 1

### B.1.1 Proof of Theorem 3.1

For notation simplicity, we define that

$$f_t(\theta) := \mathbb{E}_{z \sim P}\left[\ell_t\left(\theta, z\right)\right],$$

$$F_k(\theta) := \sum_{t=1}^{T} w_{\alpha,t}\left(\theta_k\right) \mathbb{E}_{z \sim P}\left[\ell_t\left(\theta, z\right)\right],$$

$$F(\theta) := \max_{t \in [T]} f_t(\theta),$$

where $w_{\alpha,t}\left(\theta\right) = \frac{\exp(\alpha \mathbb{E}_{z \sim P}[\ell_t(\theta,z)])}{\sum_{t'=1}^{T} \exp(\alpha \mathbb{E}_{z \sim P}[\ell_{t'}(\theta,z)])}$.

Our proof consists of two parts. The first part is to show how well $F_k(\theta_k)$ approximates $F(\theta_k)$ with our choice of the softmax hyperparameter $\alpha$. The second part is to analyze the dynamics of the algorithm and the total convergence rate.

We first show that $F_k(\theta_k) \geq F(\theta_k) + \frac{R_0 L'}{2\sqrt{k+1}}$ for $\alpha_k \geq \frac{4\sqrt{k+1}}{R_0 L'} \log \frac{4TB\sqrt{k+1}}{R_0 L'}$. Define $T_{k,\epsilon}(\theta) = \{t \in [T] \mid f_t(\theta_k) \geq F(\theta_k) - \epsilon\}$. If $\alpha_k \geq \frac{1}{\epsilon_k} \log \frac{TB}{\epsilon_k}$ for an $\epsilon_k > 0$, we have

$$
\begin{aligned}
F_k(\theta_k) &= \sum_{t=1}^{T} w_{\alpha_k, t}(\theta_k) f_t(\theta_k) \\
&\geq \sum_{t \in T_{k,\epsilon_k}(\theta_k)} w_{\alpha_k, t}(\theta_k) f_t(\theta_k) \\
&\stackrel{(A)}{\geq} \frac{\sum_{t \in T_{k,\epsilon_k}(\theta_k)} \exp(\alpha_k f_t(\theta_k))}{\sum_{t'=1}^{T} \exp(\alpha_k f_{t'}(\theta_k))} (F(\theta_k) - \epsilon_k) \\
&= \left[ 1 + \frac{\sum_{t \notin T_{k,\epsilon_k}(\theta_k)} \exp(\alpha_k f_t(\theta_k))}{\sum_{t' \in T_{k,\epsilon_k}(\theta_k)} \exp(\alpha_k f_{t'}(\theta_k))} \right]^{-1} (F(\theta_k) - \epsilon_k) \\
&\stackrel{(B)}{\geq} \left[ 1 + \frac{T \exp(\alpha_k(F(\theta_k) - \epsilon_k))}{\exp(\alpha_k F(\theta_k))} \right]^{-1} (F(\theta_k) - \epsilon_k) \\
&= [1 + T \exp(-\alpha_k \epsilon_k)]^{-1} (F(\theta_k) - \epsilon_k) \\
&\stackrel{(C)}{\geq} F(\theta_k) - 2\epsilon_k.
\end{aligned}
$$

The inequality A is due to the definition of $T_{k,\epsilon}(\theta)$. The inequality B is because $\sum_{t \notin T_{k,\epsilon_k}(\theta_k)} \exp(\alpha_k f_t(\theta_k)) \leq T \exp(\alpha_k(F(\theta_k) - \epsilon_k))$ by the definition of $T_{k,\epsilon}(\theta)$ and there exists $t_k^* \in T_{k,\epsilon_k}(\theta)$ such that $f_{t_k^*} = F(\theta_k)$, which further implies $\sum_{t' \in T_{k,\epsilon_k}(\theta_k)} \exp(\alpha_k f_{t'}(\theta_k)) \geq \exp(\alpha_k F(\theta_k))$. The inequality C is because the hyperparameter $\alpha_k \geq \frac{1}{\epsilon_k} \log \frac{TB}{\epsilon_k}$. Specifically, let $\epsilon_k = \frac{R_0 L'}{4\sqrt{k+1}}$ and $\alpha_k \geq \frac{4\sqrt{k+1}}{R_0 L'} \log \frac{4TB\sqrt{k+1}}{R_0 L'}$ correspondingly, we have $F_k(\theta_k) \geq F(\theta_k) + \frac{R_0 L'}{2\sqrt{k+1}}$ for all $k = 0, \ldots, K-1$.

We then analyze the dynamics of the algorithm and give the total convergence rate. For $k = 0, \ldots, K-1$, it holds that

$$
\begin{aligned}
F_k(\theta_k) &\stackrel{(A)}{\leq} F_k(\theta^*) + \langle \nabla F_k(\theta_k), \theta_k - \theta^* \rangle \\
&\stackrel{(B)}{=} F_k(\theta^*) + \frac{1}{2\eta} \left( \|\theta_{k+1} - \theta_k\|^2 + \|\theta_k - \theta^*\|^2 - \|\theta_{k+1} - \theta^*\|^2 \right),
\end{aligned}
\tag{14}
$$

where $\theta^* \in \arg\max_{\theta \in \Theta} F(\theta)$. The inequality A is due to the convexity of $F_k(\theta)$. The equality B is due to the update steps $\theta_{k+1} = \theta_k - \eta \nabla F_k(\theta_k)$ in Algorithm 1 and the fact $\langle a, b \rangle = \frac{1}{2}(\|a\|^2 + \|b\|^2 - \|a-b\|^2)$.

Plugging the approximation error $F_k(\theta_k) \geq F(\theta_k) + \frac{R_0 L'}{2\sqrt{k+1}}$ and the inequality $F_k(\theta) \leq F(\theta)$ for $k = 0, \ldots, K-1$ and all $\theta \in \Theta$ into (14), we have

$$
F(\theta_k) \leq F(\theta^*) + \frac{1}{2\eta} \left( \|\theta_{k+1} - \theta_k\|^2 + \|\theta_k - \theta^*\|^2 - \|\theta_{k+1} - \theta^*\|^2 \right) + \frac{R_0 L'}{2\sqrt{k+1}}.
\tag{15}
$$

Taking the average over $k = 0, \ldots, K - 1$, we further have

$$
\begin{aligned}
\frac{1}{K} \sum_{k=0}^{K-1} F(\theta_k) &\leq F(\theta^*) + \frac{1}{2\eta} \left( \sum_{k=0}^{K-1} \|\theta_{k+1} - \theta_k\|^2 + \|\theta_0 - \theta^*\|^2 - \|\theta_K - \theta^*\|^2 \right) + \frac{1}{K} \sum_{k=0}^{K-1} \frac{R_0 L'}{2\sqrt{k+1}} \\
&\leq F(\theta^*) + \frac{1}{2\eta} \left( \sum_{k=0}^{K-1} \|\theta_{k+1} - \theta_k\|^2 + \|\theta_0 - \theta^*\|^2 \right) + \frac{1}{K} \sum_{k=0}^{K-1} \frac{R_0 L'}{2\sqrt{k+1}} \\
&\overset{(A)}{\leq} F(\theta^*) + \frac{1}{2\eta} \left( \sum_{k=0}^{K-1} \|\theta_{k+1} - \theta_k\|^2 + \|\theta_0 - \theta^*\|^2 \right) + \frac{R_0 L'}{\sqrt{K}} \\
&\overset{(B)}{\leq} F(\theta^*) + \frac{K L'^2 \eta}{2} + \frac{R_0^2}{2\eta} + \frac{R_0 L'}{\sqrt{K}} \\
&\overset{(C)}{=} F(\theta^*) + \frac{2 R_0 L'}{\sqrt{K}}.
\end{aligned}
$$

(16)

The inequality A is due to the fact $\sum_{k=1}^{K} \frac{1}{\sqrt{k}} < 2\sqrt{K}$. The inequality B is because for $k = 0, \ldots, K - 1$, the function $F_k(\theta)$ is $L'$-Lipschitz continuous, which implies $\|\theta_{k+1} - \theta_k\|^2 = \eta^2 \|\nabla F_k(\theta_k)\|^2 \leq \eta^2 L'^2$. The equality C is due to our choice for the step sizes, i.e., $\eta_k = \eta = \frac{R_0}{L'\sqrt{K}}$ for all $k = 0, \ldots, K - 1$.

By the convexity of $F(\theta)$, we have $F(\bar{\theta}_K) \leq \frac{1}{K} \sum_{k=0}^{K-1} F(\theta_k)$. Combined with (16), we attain the desired result.

### B.2 Analysis for the Minimax Pre-training Method

#### B.2.1 Proof of Proposition 5.1

By the assumption on the downstream task losses $\ell_\lambda$ and the task space $\Lambda$, the equation

$$
\max_{\lambda \in \Lambda} \mathbb{E}_{z \sim P} [\ell_\lambda (\theta, z)] = \max_{t \in [T]} \mathbb{E}_{z \sim P} [\ell_t (\theta, z)]
$$

holds for all $\theta \in \Theta$.

By the definition of $\theta^*_{\max}$, we have

$$
\begin{aligned}
\max_{\lambda \in \Lambda} \mathbb{E}_{z \sim P} [\ell_\lambda (\theta^*_{\max}, z)] &= \max_{t \in [T]} \mathbb{E}_{z \sim P} [\ell_t (\theta^*_{\max}, z)] \\
&\leq \max_{t \in [T]} \mathbb{E}_{z \sim P} [\ell_t (\theta^*_{\text{average}}, z)] \\
&= \max_{\lambda \in \Lambda} \mathbb{E}_{z \sim P} [\ell_\lambda (\theta^*_{\text{average}}, z)],
\end{aligned}
$$

which is the result to prove.

#### B.2.2 Proof of Proposition 5.3

Proposition 5.3 is a standard result of gradient descent for strongly-convex and smooth functions. We include the proof here for completeness. By the convexity of $f(x)$ and the choice of the step size $\eta$ we have

$$
\begin{aligned}
f(x_{k+1}) &\leq f(x_k) + \langle \nabla f(x_k), x_{k+1} - x_k \rangle + \frac{L_f}{2} \|x_{k+1} - x_k\|^2 \\
&= f(x_k) - \left( \frac{1}{\eta} - \frac{L}{2} \right) \|x_{k+1} - x_k\|^2 \\
&\leq f(x_k),
\end{aligned}
$$

which means the objective values are nonincreasing and implies $f(x_k) \leq f(x_0)$ for all $k \in [K]$.

By the $\mu_f$-strongly convexity at the point $x^*$, it holds for all $x \in \mathbb{R}^d$ that

$$
\|x - x^*\|^2 \leq \frac{2}{\mu_f} (f(x) - f(x^*)).
$$

(17)

Combining (17) and the sequence $\{f(x_k)\}_{k=0}^K$ decreasing, we obtain

$$\|x_k - x^*\|^2 \leq \frac{2}{\mu_f} \left( f(x_0) - f(x^*) \right), \text{ for all } k = 0, 1, \ldots, K - 1,$$

as desired.

### B.2.3 Proof of Theorem 5.4

The following proposition characterizes the sample complexity to find an $\epsilon$-approximately optimal parameter by ERM for a downstream task $\lambda$ within the parameter space $\Theta_\lambda(\theta_0)$. Theorem 5.4 follows directly from Proposition B.1 by considering the worst-case downstream task $\lambda \in \Lambda$. The remaining is to prove Proposition B.1.

**Proposition B.1.** *For a given task $\lambda \in \Lambda$ and a parameter space $\Theta_\lambda(\theta_0)$, let the parameter $\hat{\theta}_\lambda^* \in \Theta_\lambda(\theta_0)$ be the minimizer in of the empirical risk for $N_\lambda$ i.i.d. samples $\{z_i\}_{i=1}^N$ from a distribution $P$, i.e., $\hat{\theta}_\lambda^* = \arg\min_{\theta \in \Theta_\lambda(\theta_0)} = \frac{1}{N_\lambda} \sum_{i=1}^{N_\lambda} \ell_\lambda(\theta, z_i)$. The parameter $\hat{\theta}_\lambda^*$ is $\epsilon$-approximately optimal with probability at least $1 - \delta$ if*

$$N_\lambda \geq \frac{8dB^2}{\epsilon^2} \log\left(1 + \frac{16L'}{\epsilon}\sqrt{\frac{2}{\mu}\mathbb{E}^*}\right) + \frac{8B^2}{\epsilon^2} \log \frac{2}{\delta}, \tag{18}$$

*where $\mathbb{E}^* = \mathbb{E}_{z \sim P}\left[\ell_\lambda(\theta_0, z)\right]$.*

Denote $\mathbb{E}_{z \sim P}\left[\ell_\lambda(\theta, z)\right]$ as $f_\lambda(\theta)$ and $\frac{1}{N}\sum_{i=1}^N \ell_\lambda(\theta, z_i)$ as $\hat{f}_\lambda(\theta)$. First, we note

$$\begin{aligned}
&\Pr\left(f_\lambda\left(\hat{\theta}_\lambda^*\right) - f_\lambda\left(\theta_\lambda^*\right) \geq \epsilon\right) \\
&= \Pr\left(\left[f_\lambda\left(\hat{\theta}_\lambda^*\right) - \hat{f}_\lambda\left(\hat{\theta}_\lambda^*\right)\right] + \left[\hat{f}_\lambda\left(\hat{\theta}_\lambda^*\right) - \hat{f}_\lambda\left(\theta_\lambda^*\right)\right] + \left[\hat{f}_\lambda\left(\theta_\lambda^*\right) - f_\lambda\left(\theta_\lambda^*\right)\right] \geq \epsilon\right) \\
&\overset{(A)}{\leq} \Pr\left(\left[f_\lambda\left(\hat{\theta}_\lambda^*\right) - \hat{f}_\lambda\left(\hat{\theta}_\lambda^*\right)\right] + \left[\hat{f}_\lambda\left(\theta_\lambda^*\right) - f_\lambda\left(\theta_\lambda^*\right)\right] \geq \epsilon\right) \\
&\leq \Pr\left(2 \sup_{\theta \in \Theta_\lambda(\theta_0)} \left|f_\lambda(\theta) - \hat{f}_\lambda(\theta)\right| \geq \epsilon\right) \\
&= \Pr\left(\sup_{\theta \in \Theta_\lambda(\theta_0)} \left|f_\lambda(\theta) - \hat{f}_\lambda(\theta)\right| \geq \frac{\epsilon}{2}\right).
\end{aligned} \tag{19}$$

The inequality A is because $\hat{f}_\lambda\left(\hat{\theta}_\lambda^*\right) - \hat{f}_\lambda\left(\theta_\lambda^*\right) \leq 0$ by the definition of $\hat{\theta}_\lambda^*$.

We derive the upper bound by covering numbers. We only consider Euclidean space for simplicity.

**Definition B.2** (Covering numbers [63, Chapter 4]). Consider a subset $S \subset \mathbb{R}^d$ and let $\epsilon > 0$. A subset $\mathcal{N} \subset S$ is called an $\epsilon$-net of $S$ if every point in $S$ is within distance $\epsilon$ of some points of $\mathcal{N}$, i.e., for all $x \in S$, there exists $x_0 \in \mathcal{N}$ such that $\|x - x_0\| \leq \epsilon$. The smallest possible cardinality of an $\epsilon$-net of $S$ is called the covering number of $S$ and is denoted $C(S, \epsilon)$, i.e.,

$$C(S, \epsilon) := \min \left\{ |\mathcal{N}| \mid \mathcal{N} \text{ is an } \epsilon\text{-net of } S \right\}.$$

Consider an $\epsilon'$-net $\mathcal{N}(\Theta_\lambda(\theta_0), \epsilon')$ of $\Theta_\lambda(\theta_0)$ where $\epsilon' = \frac{\epsilon}{8L'}$. By the definition of $\epsilon'$-net, we have

$$\sup_{\theta \in \Theta_\lambda(\theta_0)} \left|f_\lambda(\theta) - \hat{f}_\lambda(\theta)\right| \leq \sup_{\theta \in \mathcal{N}(\Theta_\lambda(\theta_0), \epsilon')} \left|f_\lambda(\theta) - \hat{f}_\lambda(\theta)\right| + \frac{\epsilon}{4}. \tag{20}$$

Combining (19) and (20), we obtain

$$\begin{aligned}
&\Pr\left(f_\lambda\left(\hat{\theta}_\lambda^*\right) - f_\lambda\left(\theta_\lambda^*\right) \geq \epsilon\right) \\
&\leq \Pr\left(\sup_{\theta \in \mathcal{N}(\Theta_\lambda(\theta_0), \epsilon')} \left|f_\lambda(\theta) - \hat{f}_\lambda(\theta)\right| \geq \frac{\epsilon}{4}\right) \\
&\leq C\left(\Theta_\lambda(\theta_0), \epsilon'\right) \sup_{\theta \in \mathcal{N}(\Theta_\lambda(\theta_0), \epsilon')} \Pr\left(\left|f_\lambda(\theta) - \hat{f}_\lambda(\theta)\right| \geq \frac{\epsilon}{4}\right)
\end{aligned} \tag{21}$$

We leverage the upper bounds of covering numbers of balls [63, Chapter 4].

**Lemma B.3.** *The covering number of a ball of radius $R$, denoted as $B_R$, in $\mathbb{R}^d$ satisfies*

$$C(B_R, \epsilon) \leq \left( \frac{2R}{\epsilon} + 1 \right)^d.$$

By Lemma B.3, we have

$$C\left(\Theta_\lambda(\theta_0), \epsilon'\right) \leq \left( \frac{2R_\lambda}{\epsilon'} + 1 \right)^d, \tag{22}$$

where $R_\lambda = \sqrt{\frac{2}{\mu} \mathbb{E}_{z \sim P}\left[\ell_\lambda(\theta_0, z)\right]}$.

By Hoeffording's inequality, for each $\theta \in \mathcal{N}(\Theta_\lambda(\theta_0), \epsilon')$, we have

$$\Pr\left( \left| f_\lambda(\theta) - \hat{f}_\lambda(\theta) \right| \geq \frac{\epsilon}{4} \right) \leq 2 \exp\left( -\frac{n\epsilon^2}{8B^2} \right) \tag{23}$$

Plugging (22) and (23) into (21), we have

$$\Pr\left( f_\lambda\left(\hat{\theta}_\lambda^*\right) - f_\lambda(\theta_\lambda^*) \geq \epsilon \right) \leq 2 \left( \frac{2R_\lambda}{\epsilon'} + 1 \right)^d \exp\left( -\frac{N_\lambda \epsilon^2}{8B^2} \right). \tag{24}$$

By (24), when the number of samples $N_\lambda$ satisfies

$$N_\lambda \geq \frac{8dB^2}{\epsilon^2} \log\left( 1 + \frac{16L'}{\epsilon} \sqrt{\frac{2}{\mu} \mathbb{E}_{z \sim P}\left[\ell_\lambda(\theta_0, z)\right]} \right) + \frac{8B^2}{\epsilon^2} \log \frac{2}{\delta},$$

we have $\Pr\left( f_\lambda\left(\hat{\theta}_\lambda^*\right) - f_\lambda(\theta_\lambda^*) \geq \epsilon \right) \leq \delta$.

## C  Example for Remark 5.2

Consider an extreme example where $\Theta = \mathbb{R}$, $\mathbb{E}_{z \sim P}\left[\ell_1(\theta, z)\right] = A(\theta - 1)^2$ where $1 < A < T$ and $\mathbb{E}_{z \sim P}\left[\ell_t(\theta, z)\right] = \theta^2$ for $t = 2, \ldots, T$. Then we have $\theta_{\max}^* = \frac{\sqrt{A}}{\sqrt{A}-1}$ and $\theta_{\text{average}}^* = \frac{A}{A+T-1}$. It further implies that

$$\ell_{\max} := \max_{\lambda \in \Lambda} \mathbb{E}_{z \sim P}\left[\ell_\lambda(\theta_{\max}^*, z)\right] = \frac{A}{\left(\sqrt{A} - 1\right)^2},$$

$$\ell_{\text{average}} := \max_{\lambda \in \Lambda} \mathbb{E}_{z \sim P}\left[\ell_\lambda(\theta_{\text{average}}^*, z)\right] = \frac{A(T-1)^2}{(A+T-1)^2}.$$

If $A = T - 1$, we have $\frac{\ell_{\text{average}}}{\ell_{\max}} \geq \frac{1}{4}\left(\sqrt{T-1} - 1\right)^2$, which is as large as $O(T)$.

## D  Training details

### D.1  Part-of-Speech Mask BERT Training Setting

In this work, we denote the number of layers (i.e., Transformer blocks) as $L$, the hidden size as $H$, and the number of self-attention heads as $A$. For comparison purposes, we primarily report results on two models with the same size: PoS-BERT$_{\text{BASE}}$ and BERT$_{\text{BASE}}$($L$=12, $H$=768, $A$=12, Total Parameters=110M). The model is trained with AdamW [42] by setting $\beta_1 = 0.9$, $\beta_2 = 0.999$, $\epsilon = $ 1e-6, and $L_2$ weight decay of 0.01. The learning rate is warmed up over the first 10K steps to a peak value of 1e-4, then linearly decayed. We duplicate training data ten times to avoid using the same mask for each training instance in every epoch so that each sequence is masked in 10 different ways over the 40 training epochs. Thus, each training sequence was seen with the same mask four times. The hyperparameters for experiments are shown as Table 4 and Table 5.

| Hyperparam | Part-of-Speech Mask BERT |
|---|---|
| Number of Layers | 12 |
| Hidden size | 768 |
| Attention heads | 12 |
| Attention heads size | 64 |
| Dropout | 0.1 |
| Warmup steps | 10K |
| Peak Learning Rate | 2e-4 |
| Batch Size | 256 |
| Weight Decay | 0.01 |
| Max Steps | 1000K |
| Learning Rate Decay | Linear |
| Adam $\epsilon$ | 1e-6 |
| Adam $\beta_1$ | 0.9 |
| Adam $\beta_2$ | 0.999 |
| Gradient Clipping | 0.0 |

Table 4: Hyperparameters for pre-training Part-of-Speech Mask BERT.

| Hyperparam | GLUE |
|---|---|
| Learning Rate | 2e-5 |
| Batch Size | 32 |
| Weight Decay | 0.1 |
| Learning Rate Decay | Linear |
| Warmup Ratio | 0.06 |

Table 5: Hyperparameters for fine-tuning Part-of-Speech Mask BERT on GLUE.

## D.2 Multi-Modal Mask MAE Training Setting

We use Vit-B [21] with a patch size of 16×16 pixels as the backbone for our MAE experiments, and estimate the model's performance under different pre-training epochs, i.e., 400 and 1,600 epochs on ImageNet1K and 800 epochs on ImageNetS50. We choose AdamW as the optimizer with a base learning rate of 1e-4 and weight decay of 0.05. We first warm up the learning rate with 40 epochs and then decay it with cosine decay [41]. We set the batch size to 2048 and trained the models using 8×A100 GPUs with automatic mixed precision enabled. Our data augmentations are straightforward. We randomly crop the images, setting the random scale between 0.2 and 1.0 and the random aspect ratio between 0.75 and 1.33. Afterward, we resize the crops to 224×224 pixels and apply a random horizontal flip with a probability of 0.5. The hyperparameters for experiments are shown as Table 6 and Table 7.

| Hyperparam | {None, GradNorm, DWA} | Uncertainty | Minimax |
|---|---|---|---|
| Batch Size | 2048 | 2048 | 2048 |
| Learning Rate | 8e-4 | 8e-4 | 8e-4 |
| Min Learning Rate | 1e-6 | 1e-6 | 1e-6 |
| Weight Decay | 0.05 | 0.05 | 0.05 |
| Adamw $\epsilon$ | 1e-8 | 1e-8 | 1e-8 |
| Adamw $\beta_1$ | 0.9 | 0.9 | 0.9 |
| Adamw $\beta_2$ | 0.95 | 0.95 | 0.95 |
| Epoch | {800} | {400, 800, 1600} | {400, 800, 1600} |
| Warm up Epoch | 40 | 40 | 40 |
| Learning Rate Schedule | cosine decay | cosine decay | cosine decay |
| Non-masked tokens | 98 | 98 | 98 |
| Input resolution | 224×224 | 224×224 | 224×224 |
| Augmentation | RandomResizeCrop | RandomResizeCrop | RandomResizeCrop |
| Dropout | 0.0 | 0.0 | 0.0 |
| Patch Size | 16 | 16 | 16 |

Table 6: Hyperparameters for pre-training Multi-Modal Mask MAE. We only pre-train 800 epochs on ImageNetS50, and pre-train both 400 and 1600 epochs on ImageNet1K.

| Hyperparam | Classification | | Semantic Segmentation | | Depth |
|---|---|---|---|---|---|
| | ImageNet1K | ImageNetS50 | ImageNetS50 | NYUv2 | NYUv2 |
| Epoch | 100 | 100 | 100 | 100 | 2000 |
| Warm up Epoch | 5 | 5 | 20 | 20 | 100 |
| Batch Size | 1024 | 1024 | 1024 | 1024 | 2048 |
| Learning Rate | 4e-3 | 4e-3 | 1e-4 | 1e-4 | 1e-4 |
| Min Learning Rate | 1e-6 | 1e-6 | 1e-6 | 1e-6 | 0 |
| Weight Decay | 0.05 | 0.05 | 0.05 | 0.05 | 1e-4 |
| Adamw $\beta_1$ | 0.9 | 0.9 | 0.9 | 0.9 | 0.9 |
| Adamw $\beta_2$ | 0.999 | 0.999 | 0.999 | 0.999 | 0.999 |
| Layer Decay | 0.65 | 0.65 | 0.75 | 0.75 | 0.75 |
| Patch Size | 16 | 16 | 16 | 16 | 16 |
| Drop path | 0.1 | 0.1 | 0.1 | 0.1 | / |
| LR Schedule | cosine decay | cosine decay | cosine decay | cosine decay | cosine decay |
| Input resolution | 224×224 | 224×224 | 224×224 | 224×224 | 256×256 |
| Augmentation | Rand(9, 0.5) | Rand(9, 0.5) | LSJ | LSJ | LSJ |

Table 7: Hyperparameters for fine-tuning Multi-Modal Mask MAE on various downtasks. The augmentation strategy LSJ is large scale jittering [26]. And we use drop path [31] in classification and semantic segmentation tasks.

# E  Limitation

Contemporary pre-training models are consistently enlarging in size. However, due to limitations associated with computational power and the non-disclosure tendency of large-scale models, we were unable to conduct our experimentation directly on ultra-large models such as LLaMA [58], GPT3 [9], and V-MoE [52]. Notwithstanding, we have validated our hypothesis on two commonly encountered domains and model frameworks, thus illustrating the extensive applicability of our proposed methodology.

