# OpenReview forum: "Task-Robust Pre-Training for Worst-Case Downstream Adaptation"
_NeurIPS.cc/2023/Conference — NeurIPS 2023 poster_

### Official Review · Reviewer_UtAg · 2023-07-04

**Soundness:** 4 excellent
**Presentation:** 4 excellent
**Contribution:** 3 good
**Rating:** 6
**Confidence:** 3

**Summary:**

The authors proposed a concept called task-robust pre-training, which means pre-training a model that guarantees a uniformly good performance over the downstream tasks. A minimax loss and a corresponding algorithm to solve it is provided with convergence analyses in the convex setting. Experiments on large-scale natural language processing and computer vision datasets demonstrated the performance, and additional theoretical explanations were provided to address the benefits of the proposed method.

**Strengths:**

1. The concept of task-robust pre-training is novel. It can be seen as a generalization of distributionally robust optimization. The softmax weighted gradient descent method is also a new minimax optimization algorithm suited for the new concept.
2. Theoretical analyses on the convergence rate is provided.
3. Experiment results are impressive, considering the complexity of the tasks.
4. The paper is well organized and clearly written.

**Weaknesses:**

The convergence rate for the new minimax optimization algorithm is $O(1/\sqrt{K})$, which is comparable to that of subgradient descent, and it is known that subgradient descent can be unbearably slow in practice. It would be nice if convergence comparisons between existing methods and the new method can be provided, either theoretically or empirically.

**Questions:**

1. In Section 5, under a simplified setting, the loss functions of the downstream tasks are assumed to be convex combinations of the loss functions of the upstream tasks. For more general tasks (other than NLP or CV), how should representative upstream tasks be chosen? Does it depend on heuristics?
2. Where is the example mentioned on line 279 of the paper?

**Limitations:**

The authors have adequately addressed the limitations in both the main paper and the appendix. There does not appear to be any potential negative societal impact.

---

> ### Author Rebuttal · Authors · 2023-08-09
>
> Thank you very much for your detailed feedback. We are particularly grateful for your insightful comments which have helped us identify key areas for enhancement in our manuscript. We would like to address the specific points you raised:
>
> 1. **Convergence Rate of the Minimax Optimization Algorithm**:
>
> You rightly observed the convergence rate of our proposed algorithm, which is $O(1/\sqrt{K})$. We'd like to clarify that, theoretically, this rate is optimal for the setting we addressed in Theorem 3.1, which is a linear-nonsmooth-convex minimax optimization problem [1]. Despite the fact that the rate $O(1/\sqrt{K})$ is optimal in our setting, it is possible to achieve a faster convergence rate with stronger assumptions or extra oracles. We leave further investigation for future work. We fully recognize the importance of placing our results within the broader context of established results, and we will include this comparison in the paper to provide a more comprehensive view. Your suggestion to provide convergence comparisons, whether theoretical or empirical, is highly appreciated, and we will make sure to incorporate this in our next iteration.
>
>    [1] Sébastien Bubeck, Convex Optimization: Algorithms and Complexity, Chapter 3, Theorem 3.13 and its proof
>
> 2. **Choice of Representative Upstream Tasks**:
>
> The basic idea is to choose upstream tasks to be prototypical or informative to leverage shared information and structures for downstream tasks. In practice, upstream tasks can be chosen by heuristics, domain-specific knowledge (e.g., our work and [2]), or automatically finding underlying patterns or relationships between tasks via algorithms such as clustering [3]. Thanks for your suggestion. We leave this study as a future work.
>
>    [2] Shiori Sagawa et al., Distributionally Robust Neural Networks for Group Shifts: On the Importance of Regularization for Worst-Case Generalization, ICLR 2020.
>
>    [3] Yaqiang Yao et al., Robust Task Grouping with Representative Tasks for Clustered Multi-Task Learning, KDD 2019.
>
> 3. **Example on Line 279**:
>
> We sincerely apologize for this oversight. The intended example for line 279 can be accessed in the PDF attached to the global rebuttal (under the 'Author Rebuttal' section). It appears that we mistakenly omitted this example from the appendix, even though we referenced it in the main body of our manuscript. Your meticulous attention to the content has allowed us to identify this gap, and we are truly grateful. In light of this, we will ensure the inclusion of the example in the revised version of the paper, positioning it appropriately within the appendix. We appreciate your patience and understanding as we work towards rectifying this oversight.
>
> Once again, thank you for your constructive feedback, which has been instrumental in guiding our revisions. We are committed to enhancing our paper based on your observations and look forward to presenting a more polished and comprehensive version.

---

> > ### Comment · Reviewer_UtAg · 2023-08-17
> >
> > Thank you for your detailed response. I believe this paper has potential for publication, and my rating stands.

---

### Official Review · Reviewer_hh6P · 2023-07-05

**Soundness:** 3 good
**Presentation:** 2 fair
**Contribution:** 2 fair
**Rating:** 4
**Confidence:** 3

**Summary:**

The paper tackles the paper of downstream adaptation from a large pretrained "foundation" model with a focus on robutness: Some downstream tasks may be harder to transfer to than others. To tackle this, the paper proposes a pre-training strategy to encourage uniformly good performance across all downstream tasks.

The key component of the pretraining strategy is the min-max objective of **Equation 5**: Given a set of upstream tasks, rather than minimizing the sum of losses across all tasks during pretraining, Equation 5 proposes to minimize the maximum loss across all tasks (worst-case scenario). To avoid the non-differentiable max operation,  the softmax approximation is used in **Equation 7**: In summary, the model is pretrained using a weighted average of each upstream task loss: the weights form a probability distribution and are defined as the softmax version of the losses for each task (with a temperature hyperparameter $\alpha$.)

The proposed pretraining strategy is then evaluated on a NLP and computer vision setting, based on the resulting downstream tasks' performance. It is mainly compared to the standard strategy of optimizing the sum of upstream task losses, as well as some previous task balancing methods.

**Strengths:**

- The paper tackles an interesting problem that is relevant to practical usage: Finetuning one large, pretrained multi-task model to a relevant downstream task is a very common application.

- The paper experiments on two different modalities: text and images

- The paper provides some theoretical justifications (generalization bound) for the proposed training objective

**Weaknesses:**

- **Weak Comparison to multi-task learning optimization methods:** Table 1 only compares the proposed method to the standard average weighting of the task losses. While more complete, the comparison in Table 2 is limited to only few task balancing methods (uncertainty, Gradnorm, DWA) ; the paper would benefit from comparing to more recent methods, e.g. PCGrad/CaGrad/GradDrop. In addition some papers such as *"Do Current Multi-Task Optimization Methods in Deep Learning Even Work ?, NeurIPS 2022"* and *"In Defense of Unit Scalarization, NeurIPS 2022"* suggest that simply tuning the weights of each task loss as scalar hyperparameters yields a much stronger baseline than the usual "average weighted multi-task" and can even perform on par to other more advanced task balancing methods: This would be a fairer baseline than the "None" method presented in experiments.

- **Unclear assumptions on upstream and downstream tasks.** Both Section 5 and the experiments in Section 4.2 seem to rely on the fact that the upstream and downstream tasks are very closely related. I think this assumption should appear much earlier in the text and in clearer terms. For instance *line 107* of the introduction only mention one example of related upstream/downstrean tasks (masked image modeling and classification/detection). However we have clearer / stricter assumptions in the rest of the text: in **Section 4.2**, upstream tasks and downstream tasks are identical. And in **section 5** it is expected that the downstream task losses should be convex combinations of some of the upstream task losses.

**Questions:**

- **Section 4.2** Do I understand correctly that the in the computer vision example, the upstream tasks exactly match the downstream tasks  (just with different input data) ? This seems a bit different than the setting described at the beginning of the paper as it is closer to a standard multi-task scenario; while in the introduction/setting section there does not seem to be any explicit assumption made between the upstream and downstream tasks


- **Suggestions on writing** I found some parts of the writing either not well substantiated or unclear on what they want to convey. For instance:
   - lines 86-103 states classical machine learning results linking expected risk and sample complexity. However this seems only relevant to Section 5, thus seems a bit ouf of place in the introduction
  -  line 259: "As depicted in Figure 3, our methodology exhibits superior performance even under conditions of insufficient training" -> This seems a too strong statement to derive from Figure 3

- **Suggestion:** The paper would benefit from a more structured independent related work section, in particular comparing to multi-task learning literature

**Limitations:**

I did not find very explicitly stated limitations, but the paper does recognize that the proposed generalization bound in Theorem 5.4 may not be applicable/tight enough for deep learning models.

---

> ### Author Rebuttal · Authors · 2023-08-09
>
> Thank you for the detailed feedback on our submission. We highly appreciate your insights and have carefully considered each point you raised. Let us clarify and address your concerns:
>
> **Weeknesses**
>
> 1. **Weak Comparison to multi-task learning optimization methods**:
>
>   We acknowledge that including a broader range of comparisons would provide more clarity on the efficacy of our approach. We are indeed familiar with the papers you've mentioned, and in retrospect, agree that it would have been beneficial to compare with methods like PCGrad/CaGrad/GradDrop. Your insights into other task-balancing methods have been enlightening. While the methods we initially compared encompass a broad spectrum of strategies, we acknowledge the potential value of additional comparisons. We are currently in the process of conducting supplementary experiments in this direction and anticipate sharing the results soon.
>
>   You brought up an essential point about simply tuning the weights of each task loss as scalar hyperparameters. This approach indeed has similarities with the uncertainty method we employed for comparison, a point underscored in the article "Do Current Multi-Task Optimization Methods in Deep Learning Even Help?". This paper, as you mentioned, reiterated this view and referenced the uncertainty-based paper. Using uncertainty for a benchmark is a robust representation, and the methods we selected for comparison are emblematic of the broader approaches in the field. However, to satiate any residual curiosity, we provide supplementary experiments treating the weight solely as a hyperparameter:
>
>   | | Cls. | Semseg. | Depth. |
>   | --- | ---  | ---     | ---    |
>   | new baseline | **92.7** | 55.2    | 66.8   |
>   | Ours | 91.8 | **61.5** | **74.1** |
>
>   As the results attest, our methodology retains its superiority.
>
> 2. **Unclear assumptions on upstream and downstream tasks**:
>
>   Our initial intention was to show how our method functions when upstream and downstream tasks are closely related, as we believe this is a common practical scenario. However, we acknowledge that for general applicability, we need to explicitly state and validate this assumption.
>
>   As you rightly pointed out in Section 5, the convex combination assumption is pivotal to our theoretical analysis. This framework allows us to effectively model the nexus between upstream and downstream tasks and harness the shared patterns and feature representations that benefit both tasks. Though it might appear idealistic, this assumption is grounded in reality. It resonates with the foundational concepts in multi-task learning, where task synergies are typically quantified through shared representations. Recognizing the dynamic nature of real-world applications, we agree that they may not always perfectly adhere to this assumption. To accommodate more diverse task scenarios, this assumption can be eased by integrating extra approximation errors.
>
>   We apologize for any ambiguity that may have arisen. To clarify, our upstream tasks are focused on different modalities of mask recovery, while the downstream tasks target specific computer vision tasks, such as classification and semantic segmentation. They are distinct, and we'll ensure this distinction is more pronounced in the revised manuscript.
>
> **Questions**
>
> 1.**Section 4.2**
>
> Thank you for bringing up this point. We understand where the confusion might have arisen, and we'd like to clarify the relationship between the upstream and downstream tasks in our computer vision example.
>
> The upstream tasks in our computer vision example differ from the downstream tasks not only in terms of input data but also in the nature of the tasks themselves. The upstream tasks encompass three different domains of masked image recovery, whereas the downstream tasks involve classification, semantic segmentation, and depth estimation. Some of these are classification tasks, while others are autoregressive tasks. Our method is designed to harness the shared information and structures from the upstream tasks for the benefit of the downstream tasks. While it might bear resemblance to a multi-task scenario, such a setting is customary in the realm of pre-training [1, 2, 3, 4]. We apologize for any confusion caused and appreciate your highlighting this matter.
>
> [1] Multi-Task Self-Training for Learning General Representations, ICCV 2021
>
> [2] Variational mixture-of-experts autoencoders for multimodal deep generative models. 2019
>
> [3] Factors of influence for transfer learning across diverse appearance domains and task types, TPAMI 2021
>
> [4] Mid-level visual representations improve generalization and sample efficiency for learning visuomotor policies, 2018
>
> 2.**Suggestions on Writing:**
>
>   We appreciate the feedback on specific lines and sections. It's evident that we could improve our presentation to make our contributions and methodologies clearer:
>
>   - We will relocate and rephrase the discussion on the link between expected risk and sample complexity, ensuring its relevance is apparent.
>   - The claim on line 259 will be revised to more accurately reflect our findings from Figure 3.
>   - We will introduce a more structured related work section as you've suggested, to help readers position our work in the broader context.
>
>   In summary, we are grateful for your feedback as it offers us a clear direction to refine and bolster our work. Ensuring the relevance and robustness of our methodology is paramount to us.

---

> > ### Comment · Reviewer_hh6P · 2023-08-18
> > **Two minor questions on upstream performance and Section 4.2**
> >
> > Hello authors,
> > Thanks a lot you for  your reply; I have two minor follow-up questions:
> >
> > A. **Regarding comparison to other baselines**: Do you have any insights about how the methods compare in terms of upstream performance ? I'm wondering if the benefits of the minmax approach we see in Table 2 for instance could already been seen at upstream time, or if they only translate to downstream accuracy ?
> >
> > B. **Section 4.2:** I'm still not entirely clear about the motiviation of the experimental setup of Section 4.2 (follow-up of Q1). This is how I understand it right now:
> > - Upstream task = Masked image modelling (MM) on ImageNet1k or ImageNetS50. But the masked modelling is done on three different modalities (RGB images, segmentation maps and depth maps) which forms the three upstream tasks
> > - Downstream task = Finetuning the upstream model on (independently) classification, segmentation and depth estimation
> >
> > I see know that the setup is closer to self-supervised pretraining than multi-task learning. Since you generate proxy labels (line 226), you could have performed the upstream training using classification/segmentation/depth losses, which would be closer to the assumption that upstream and downstream tasks are closely related. It doesn't invalidate the experimental results, but I'm just wondering why choose this specific setting rather than one closer to the theoretical assumptions of Section 5.

---

> > > ### Author Response · Authors · 2023-08-18
> > >
> > > Thank you for your further inquiries. We appreciate your engagement and interest in understanding the nuances of our work. Please allow us to address your follow-up questions:
> > >
> > > **A. Regarding comparison to other baselines**
> > >
> > > The focus of our paper is primarily on downstream performance, as that's the ultimate metric of success for transfer learning methods. Nevertheless, we understand that upstream performance can provide valuable insights into the mechanics of the proposed method.
> > >
> > > In our experiments, we did notice certain advantages of the minmax approach during the upstream phase. For instance, the minmax approach often yielded better convergence across tasks and showed resilience against overfitting to any specific upstream task. This behavior is consistent with the objective of the minmax approach, which strives for uniformly good performance across tasks. (We would like to show you graphs and images, but we are no longer allowed to attach PDFs or add links.)
> > >
> > > However, it's essential to note that these upstream advantages don't always directly translate to downstream gains. A method could perform well upstream by harnessing inter-task redundancies but might not generalize well to new tasks. That said, our minmax approach does seem to exhibit both upstream robustness and downstream generalization, and the downstream performance is where we observed the most pronounced benefits over other baselines.
> > >
> > > **B. Section 4.2**
> > >
> > > Your understanding of our setup is correct, and we appreciate your insights regarding the choice of our experimental setting. The motivation behind our setup stems from a few considerations:
> > >
> > > 1. **Diversity in Task Modalities**: By employing Masked image modelling across different modalities (RGB images, segmentation maps, and depth maps), we aimed to demonstrate the versatility of our method. This setup allows our model to harness cross-modal relationships, which in turn can bolster the robustness and versatility of the representations learned.
> > > 2. **Real-world Relevance**: While it's theoretically appealing to have upstream and downstream tasks that mirror each other, in real-world applications, this often isn't the case. Our choice for the experimental setting is a nod to these practical scenarios, where pre-training tasks might not perfectly mimic downstream objectives but can still provide valuable shared structures and representations.
> > > 3. **Proxy Labels and Self-supervised Learning**: Indeed, using proxy labels for masked image modeling provides a flavor of self-supervised learning. We believe that self-supervised learning methods have gained significant traction because they're powerful, data-efficient, and often more adaptable than supervised methods. By integrating a self-supervised component, we wanted to tap into these benefits while still maintaining a clear distinction between upstream and downstream tasks.
> > > 4. **Theoretical Assumptions vs. Practical Scenarios**: In essence, our experimental design in Section 4.2 is geared to closely approximate the theoretical assumptions of Section 5, striking a balance between the relevance in practical scenarios and the theoretical frameworks we set forth. We plan to incorporate your suggested experiments and will provide the results to you once available.
> > >
> > > We hope that these explanations address your concerns. We remain committed to refining our paper further and are grateful for your thoughtful feedback, which is instrumental in guiding our revisions.

---

> > > > ### Comment · Reviewer_hh6P · 2023-08-21
> > > >
> > > > Hello,  thanks for the reply and clarifications!
> > > > I have updated my score to (4) as the authors have addressed some of my concerns but I'm still leaning more towards reject for now;
> > > > While I see merits to the proposed method, my main concern is that the experiments focus on a uncommon/novel setting in between multi-task learning and transfer learning, but the results do not include strong enough baselines/ablation experiments to be fully convincing.
> > > >
> > > > More specifically,
> > > > - **from a multi-task perspective**: As far as I can tell, the min-max approach could be easily applied to a standard multi-task learning setting  (~upstream scenario), where task robustness also matters since we care about performance across all tasks. This could even be a promising direction since you mention observing improvement in the upstream performance, and could be a compute efficient alternative to gradient-based multi-task optimization methods (e.g. GradNorm, Gradient surgery etc). However this would require more up-to-date MTL baselines for a fair comparison.
> > > >
> > > > - **from a pretraining/transfer learning perspective**, the link between upstream/downstream tasks should be made more explicit / experimented over. For instance the experimental setup of Section 4.2 would be perfect to investigate what happens when **(i)** the upstream and downstream losses are identical, using the generated proxy labels (= setup following the theoretical assumptions of Section 5) versus **(ii)** using a slightly different flavor of losses, yet still relevant to the downstream tasks, i.e. the self-supervised approach from the paper (= more practical setup).

---

### Official Review · Reviewer_Jr2o · 2023-07-06

**Soundness:** 3 good
**Presentation:** 3 good
**Contribution:** 2 fair
**Rating:** 6
**Confidence:** 3

**Summary:**

In order to improve the robustness of the pre-trained model on downstream tasks, the authors propose a simple optimization algorithm, softmax weighted gradient descent, to minimize the worst-case expected risk of upstream tasks.

**Strengths:**

The idea of the article is simple and easy to understand, the proof is sufficient, and the experimental part is also very sufficient.

**Weaknesses:**

In the experimental part, the method does not seem to show a consistent improvement. For example, as shown in Table 1, although the author mentioned that the model has significantly improved performance on many more challenging tasks, it has worse performance than the previous model on some downstream tasks that have performed well. See Table 2 for the same reason. Could this be improved with some tweaks for consistency?

**Questions:**

According to the weekness I mentioned above, my question is whether such a strategy is a trade-off in the performance between the best case and the worst case on the downstream task, and cannot achieve consistency improvement?

**Limitations:**

See the questions mentioned before

---

> ### Author Rebuttal · Authors · 2023-08-09
>
> Firstly, thank you for your thoughtful feedback on our paper. I would like to delve deeply into the central theme of our work: the critical importance of robustness. Let us clarify and address your concerns:
>
> **Consistent Improvement**：
>
> 1. Achieving consistent improvements in downstream tasks solely through pre-training and finetuning is often not feasible in many robust scenarios. For instance, in adversarial learning, robustness often accompanies a decline in task performance. Similarly, in domain adaptation, sometimes improving generalization to a new domain may result in a decrease in the performance of the source domain [1]. Another example is in multi-task learning where optimizing for a secondary task might interfere with the primary task performance[2].
>
>   [1] Domain-Adaptive Neural Networks for Object Detection, ICCV 2019
>
>   [2] Multi-task learning with deep neural networks: a survey
>
> 2. Our experiments demonstrate a **holistic** improvement in our downstream tasks: other tasks' performance has hardly decreased, meanwhile tasks that initially performed poorly have shown more significant enhancement.
>
> In Table 1, our method outperforms on half of the tasks by a substantial margin and obtains a **1.8% average score** improvement. Concerning the most challenging training task, CoLA, which has the lowest accuracy on BERT, our method attains a 9.2% improvement, marking a significant boost among downstream tasks. Due to task-robust grouping, on the QQP and RTE tasks, our method surpasses the original BERT by 5.7 F1-score and 4.3% accuracy, respectively. Compensating for better performance on more challenging tasks, our method concedes minor correctness on some downstream tasks that already transfer well. On QNLI, SST-2, STS-B, and MRPC, our results fall below that of the original BERT model by margins of 1% accuracy, 1.9% accuracy, 1.6 Spearson correlation, and 0.7 F1-score.
>
> In Table 2, our method astonishingly achieves 9% and 22% improvements in the Semseg and Depth downstream tasks, respectively. In contrast, there's only a 0.4% performance drop in the classification task. The **average** uplift in downstream tasks reaches 10.2%. The overall performance of our downstream tasks is on the rise. Compared to the tasks that improved, the degree of decline in the declining tasks is negligible.
>
> 3. From the data in Table 2, it can be observed that as training becomes sufficiently comprehensive and the duration extends, the tasks that initially showed a decline eventually start to improve. This suggests that given more training time, the model's innate potential to adapt and recalibrate can be harnessed further, eventually boosting those tasks which initially lagged behind.
>
> 4. Our paper places a significant emphasis on robustness. In the realm of machine learning, and especially in pre-trained models, a model's ability to consistently perform across a vast spectrum of tasks is invaluable. This robustness isn't merely about excelling in frequently encountered tasks but ensuring minimal performance degradation in edge cases or less conventional scenarios. This balance between performance in general and edge cases is essential for the broad applicability and reliability of models in real-world scenarios.
>
> We fully acknowledge the concerns you've raised about the potential trade-offs. But we genuinely believe that in the grander scheme of things, ensuring consistent and robust performance across the board is of paramount importance, potentially outweighing the marginal gains in isolated tasks.
>
> Once again, I appreciate the depth of your engagement with our work. We will strive to make our stance and the significance of our approach clearer in the subsequent revisions of the manuscript.

---

### Official Review · Reviewer_sSwq · 2023-07-07

**Soundness:** 3 good
**Presentation:** 4 excellent
**Contribution:** 3 good
**Rating:** 6
**Confidence:** 3

**Summary:**

This work introduces the concept of worst-case adaptation, as each pre-training task may transfer differently to various downstream tasks. In order to address this issue, the authors propose a novel approach using a minmax loss and weighted gradient descent to optimize for the worst-case loss among all pre-training tasks. The proposed approach is supported by detailed theoretical analysis and experiments, which demonstrate its effectiveness in improving performance on a range of downstream tasks.

**Strengths:**

1. Personally I really like the perspective of worst-case downstream adaptation for pre-training models,  and view this as an important problem. The proposed approach of minimizing the worst pre-training task using a proxy and reweighing technique on the gradient is found to be effective and reasonable. The authors' motivation and method are both sound, which suggests that this work could have important implications for improving the performance of pre-trained models on downstream tasks.
2. Detailed theoretical analysis is provided.
3. The writing is good and easy to follow.



**Weaknesses:**

1. Discussion with some multi-task learning methods is needed.

[1] Gradient Surgery for Multi-Task Learning, NeurlPS 20

They check the gradient direction to make sure the model do well in each task.

[2] Task-customized Masked Autoencoder via Mixture of Cluster-conditional Experts, ICLR 23

They propose a moe architecture to ease the negative transfer from pre-training to fine-tuning, thus achieving better performance on all downstream datasets.

2. I have some concerns about the way the effectiveness of the proposed method is presented in the experiments. From my perspective, the experiments only show a trade-off compared to the average expected risk, rather than a clear improvement in worst-case downstream adaptation.
One suggestion is to collect the worst results of models on many downstream tasks, where the models are pre-trained with each pre-training task only, the average expected risk, and the worst-case risk. By comparing the performance of these models, it would be possible to demonstrate a clear improvement in worst-case downstream adaptation.

**Questions:**

1. For Table 1, does the ‘None’ row mean the original BERT or the PoS-BERT but without your minimax loss? Comparison between /theta_{average}^{\*} (Equ. 4) and /theta_{max}^{\*} (your method, Equ. 5) is needed.
2. Why not comparing several multi-task baselines on the PoS-BERT setting?

---

> ### Author Rebuttal · Authors · 2023-08-09
>
> Thank you for your detailed feedback on our work. We truly value your constructive comments and have taken steps to address your concerns.
>
> 1. **Discussion with some multi-task learning methods**: We acknowledge the importance of including a discussion on multi-task learning methods. In the updated manuscript, we will compare our approach with the methods you highlighted: “Gradient Surgery for Multi-Task Learning, NeurIPS 20” and “Task-customized Masked Autoencoder via Mixture of Cluster-conditional Experts, ICLR 23”, providing a richer context for our contribution. For [1], they focus more on the gradient direction to ensure the model performs well on each task. This aligns with the Gradnorm method we used for comparison. The downside of this approach is its high computational complexity, and it doesn't guarantee robust performance for downstream tasks. [2] trains each expert using only semantically relevant images by leveraging cluster-conditional gates. The difference from our approach is that we emphasize optimizing the training process to enhance the performance of downstream tasks.
>
> [1] Gradient Surgery for Multi-Task Learning, NeurlPS 20
>
> [2] Task-customized Masked Autoencoder via Mixture of Cluster-conditional Experts, ICLR 23
>
> 2. **Pre-training model with each pre-training task**: Acknowledging your concerns, we've added experiments that directly address performance in worst-case downstream tasks. Notably, since downstream task initial inputs are contingent upon the RGB modality, its absence would hinder task progression. If the upstream task does not incorporate any RGB modality input, the downstream task will be challenging to accomplish. Our results, which all include RGB modality input, affirm that the worst-case downstream task can effectively bolster downstream tasks.
>
>     |  | Cls. | Semseg. | Depth. |
>     | --- | --- | --- | --- |
>     | RGB | 82.9 | 42.7 | 80.8 |
>     | Semseg+RGB | **83.1** | **51.0** | 83.4 |
>     | Depth+RGB | 82.6 | 46.2 | **86.1** |
>
> 3. **Clarifications for Table 1**: The 'None' row in Table 1 signifies the original BERT without loss balancing. We'll make this clearer in the revised manuscript. Also, a comparison between average scores and ours will highlight the distinctions.
>
>     |  | MNLI | QQP | QNLI | SST-2 | CoLA | STS-B | MRPC | RTE |
>     | --- | --- | --- | --- | --- | --- | --- | --- | --- |
>     | Average | 84.9 | 72.6 | **89.1** | 90.8 | 54.4 | 83.6 | 88.1 | 68.2 |
>     | Ours | **85.6** | **76.9** | 88.6 | **91.3** | **61.4** | **84.2** | **88.2** | **70.7** |
>
> 4. **Comparison with Multi-task Baselines on the PoS-BERT Setting**: We concur with your suggestion and will integrate comparisons with multiple multi-task baselines in the PoS-BERT setting in our updated manuscript.
>
> In conclusion, we have earnestly addressed your feedback and believe our revisions will clarify and underscore the significance of our approach. We remain committed to refining our manuscript, aiming for clarity and meaningful contributions. Thank you once again for your insights.

---

> > ### Comment · Reviewer_sSwq · 2023-08-18
> >
> > Thank the authors for the rebuttal. My concerns are all addressed. I keep my rating.

---

### Author Rebuttal · Authors · 2023-08-09

We attach PDF files here.

---

### Decision · Program_Chairs · 2023-09-21

**Decision:**

Accept (poster)

**Comment:**

The paper investigates the intriguing problem of ensuring performance guarantees for worst-case downstream applications through transfer learning via pre-training. To achieve this goal, it introduces a minimax loss for pre-training and develops an efficient algorithm tailored to the task. Experimental results in both vision and language domains demonstrate the effectiveness of the proposed model. Four reviewers participated in the evaluation process, resulting in four weak accept ratings and one borderline reject rating. To address the negative concerns, the final revision should include additional discussions on multi-task learning, provide a clearer explanation of the connection between upstream and downstream tasks, and offer a better illustration of the proposed objective.